# Laser Transmission Characteristics of Seawater for Underwater Wireless Optical Communication

**DOI:** 10.3390/s25103057

**Published:** 2025-05-12

**Authors:** Ruiman Yuan, Tinglu Zhang, Cong Li, Hong Gao, Lianbo Hu

**Affiliations:** 1College of Marine Technology, Ocean University of China, Qingdao 266100, China; m17854296271@163.com (R.Y.); gaohong@stu.ouc.edu.cn (H.G.); hulb@ouc.edu.cn (L.H.); 2Laboratory for Regional Oceanography and Numerical Modeling, Qingdao Marine Science and Technology Center, Qingdao 266237, China; 3Institute of Telecommunication and Navigation Satellites, China Academy of Space Technology, Beijing 100086, China; liccast@163.com

**Keywords:** underwater optical wireless communication, laser transmission, blue-green laser, Monte Carlo simulation

## Abstract

Channel modeling of seawater is essential for understanding the transmission process of underwater laser light and optimizing the system design of underwater wireless laser communication. This study systematically examined the transmission characteristics of underwater blue-green laser communication, such as the angle of arrival, beam spreading, and channel loss, based on the Monte Carlo ray tracing method, across three different waters. The statistical analysis has led to the following definitive conclusions: (a) The differences in average AOA are profound in clear water and at short attenuation lengths in coastal and turbid harbor waters and are small at long attenuation lengths. The differences in average AOA between the offsets of 0 m and 10 m are about 62.3° and 12.9° at the attenuation lengths of 1 and 25 in clear water. The differences between offsets of 0 m and 10 m in average AOAs are about 74.4° and 5.8° in coastal water and 67.2° and 12.2° in turbid harbor water at the attenuation lengths of 1, 20, and 35, respectively. (b) The beam diameters are 0.1 m at the attenuation length of 25 in clear water and 83.8 m and 25.3 m when the attenuation length is 35 in coastal and turbid harbor waters. It manifests that the beam spreading is indistinctive in clear water while prominent in coastal and turbid harbor waters. (c) The difference in the received power at the various offsets decreases with increasing attenuation length but with distinct patterns. Take the offsets of 0 m and 10 m as examples. The absolute difference in the power loss reduces from 88.0 dB·m^−2^ to 46.8 dB·m^−2^ when the attenuation length reaches 25 in clear water. At the attenuation lengths of 1 and 35, the power losses are 94.9 dB·m^−2^ and 4.3 dB·m^−2^ in coastal water and 117.4 dB·m^−2^ and 12.6 dB·m^−2^ in turbid harbor water. Moreover, the minimum underestimation of power loss by applying Beer’s Law could be almost 2 dB·m^−2^ in turbid harbor waters. To achieve a high receiving gain, the weighted average angles of arrival at different offsets indicate that a small field of view is advantageous in clear water and at short transmission distances in coastal and turbid harbor waters. In contrast, a larger field of view is effective at long transmission distances in coastal and turbid harbor waters. Additionally, the absolute differences in channel losses at various offsets suggest that alignment between the transmitter and the receiver is crucial in clear water and at short transmission distances in coastal and turbid harbor waters. In contrast, misalignment may not lead to significant channel loss at longer transmission distances in turbid harbor water. The results of this study underscore the importance of considering water type, transmission distance, and offsets relative to the beam center when selecting receiver parameters.

## 1. Introduction

As more and more underwater human activities develop, such as environmental monitoring, underwater scientific data collection, marine archaeology, offshore oil field exploration, port security, and marine military, the demand for underwater communication is rapidly increasing [1]. Wired communication systems, such as fiber optic systems, can provide real-time data transfer underwater but have a limited scope in practical applications because of high cost and operational difficulty. Thus, underwater wireless links emerge to meet the actual needs. Acoustic communication technology is commonly used for underwater wireless communication, offering a communication range of over several kilometers underwater with a low bit error rate (BER). However, it typically provides low data transmission rates around tens of kbps, which are insufficient for high-bandwidth applications such as image and real-time video transmission [2]. In contrast, underwater wireless optical communication meets the demands of short- to medium-distance applications and could offer a significantly higher transmission rate [3]. Due to the lower absorption of the blue-green laser within 450~550 nm underwater [4], laser diodes (LDs) or light-emitting diodes (LEDs) could be used in underwater optical communication (UWOC) systems to achieve in the order of tens of Mbps data transfer rate [5,6] and exceeding Gbps in short or medium-long experimental tanks [7,8].

In recent years, numerous studies have focused on underwater laser communication regarding channel modeling [9,10,11,12,13,14] and system design and development [15,16,17,18,19,20,21], among which channel modeling is crucial for understanding how underwater laser light is transmitted and optimizing system design. The BER is a vital parameter of the UWOC system and is relevant to the signal-to-noise ratio (SNR) [22]. Therefore, when designing an underwater laser communication system, the key challenge is maximizing the SNR, which is proportional to the optical power received by the receiver. The received power after a certain distance can be expressed as [23]:(1)$$P_{R}(t,d,r)=P_{T}(t)G_{T}P_{Lw}(t,d,r)G_{R}$$
where $P_{R}\left(t,d,r\right)$ is the received power, which is a function of the transmission time t, the offset relative to the emitted laser beam center d, and the transmission distance r, $P_{T}(t)$ is the transmitting power, $G_{T}$ is the transmitter gain determined by the beam aperture and divergence, $P_{Lw}(t,d,r)$ is the channel loss due to absorption and scattering, and $G_{R}$ is the receiver gain determined by the receiver aperture and field of view (FOV). In addition to the absorption by the water medium, underwater channel loss $P_{Lw}\left(t,d,r\right)$ is caused by spatial and temporal dispersion, which is attributed to the scattering of water molecules and particles as well as the natural divergence of the emitting laser. Due to multiple scattering events, spatial dispersion temporal dispersion, and furtherly constrain the signal bandwidth of underwater laser channels. In brief, the underwater signal intensity is determined by the optical properties of the water bodies.

There are two main methods for channel modeling: approximate analytical method [24] and numerical method [25]. The approximate analytical method gives a closed form established under certain approximate conditions, resulting in a lack of accuracy and further restriction of the application scope. In contrast, the Monte Carlo simulation method is a more practical way since it can handle various complex nonlinear problems without simplifying or making any assumptions. The main thread of Monte Carlo ray tracing [14,26,27,28,29,30] is recording photon trajectories by simulating the photon’s interactions with the optically active substance in water based on sampling from the probability density functions (PDFs) associated with the water’s inherent optical properties (IOPs). For a given number of photons and IOPs, the received power and associated transmission distance of each photon are computed to obtain the channel impulse response.

Primary topics of research regarding laser transmission characteristics include power loss [14,27,28,31], laser beam spreading [26,27,32], angle of arrival (AOA) distribution [33], and temporal dispersion [14,30]. Qadar et al. [28] analyzed the power loss and BER of the three typical waters while considering the receiver aperture and FOV modulation; however, this study is solely for SISO-based UWOC links. Elamassie et al. [9] analyzed the power loss and BER of a large divergence angle light source transmitted in four typical waters based on Monte Carlo simulation and is instructive for the deployment of relay-assisted UVLC systems. Cox and Muth [32] presented the received power loss in three typical waters and three harbor waters; the simulation was validated with water tank experimental results conducted by Cochenour [23]. Zhang et al. [33] derived a closed-form expression of the AOA distribution in three typical waters by assuming ballistic and single scattering conditions, but the accuracy of this single scattering assumption is limited to long-distance transmission and turbid waters [34].

In summary, there is still a lack of systematic analysis of the spatial dispersion characteristics of UWOC. To deal with various application scenarios, we used the Monte Carlo ray tracing method to systematically analyze the spatial dispersion characteristics of laser transmission, including beam spreading, AOA, and channel loss at different transmission distances for typical waters. The IOPs of a water body could be crucial parameters for insight into the underwater laser transmission features. In this study, we considered the linear misalignment of the receiver by calculating the power loss at different offsets on the receiving plane; thus, this result could be applied to the transmission characteristics analysis of SISO [35] and SIMO systems [36].

In the presence of these considerations, this paper is organized as follows. Section 2 introduces Monte Carlo ray tracing technology and simulation settings including emitted laser beam width, FOV, receiver conditions, and optical properties of three typical waters. Section 3 discusses the simulated results, including the analysis of the AOA distribution, beam spreading, and channel loss, to draw a whole impression of the laser spatial dispersion. Section 4 shows the overall analysis and summary of this article.

## 2. Method

In this study, we analyzed the transmission characteristics of an underwater laser based on the three-dimensional Monte Carlo simulation program by providing the typical parameters of the emitted laser and the optical properties of three typical waters (i.e., clear water, coastal water, and turbid harbor water). The abbreviations and symbols used are listed in Table 1, the majority of which are the parameters in the marine optics area [37]. The simulations were conducted using MATLAB R2019b.

### 2.1. Monte Carlo Simulation

To analyze the spatial dispersion characteristics of laser transmission, this study used Monte Carlo photon tracking technology to obtain the underwater light field distribution [38]. This method enables the tracking of photons from the light source to the receiver plane. The fundamental concept involves sampling variables with known probability distributions, which include the transmission distance of the photons, the interaction type between the photon and the medium, and the scattering direction. By calculating the trajectories of the transmitted photons that reach the receiving plane, it is possible to further analyze the relevant information related to the light field.

Sampling from the cumulative distribution function Ψ(x) is based on the following basic rule:(2)$$\zeta =\int_{x_{min}}^{x_{max}}\;p\left(x\right)dx=\Psi (x)$$
where ζ is a uniform random variable distributed between 0 and 1, x is the sampling parameter, $x_{min}$ and $x_{max}$ are the minimum and the maximum values of x, p(x) is the probability density distribution function, $\int_{-\infty }^{+\infty }\;p\left(x\right)dx=1$.

#### 2.1.1. Photon Transmission Distance

Determine the transmission distance of photons before the next interaction with the medium. The attenuation law of a collimated beam with radiance $L(r,\vec{\xi })$ with distance is [39](3)$$L(r,\vec{\xi })=L(0,\vec{\xi })\cdot e^{-cr}$$
where r is the geometric transmission distance of the photon in the direction $\vec{\xi }$, and r is 0 in the reference point.

The mean free path 1/c is the average distance traveled by a photon between two interactions. Therefore, the probability of a photon interacting within a length dr is c·dr, and the probability of no interaction occurring is 1−c·dr. If the transmission distance r is divided into *N* segments equally, the probability of no interaction occurring within the transmission distance r is(4)$$\Psi (r)={\left(1-\frac{cr}{N}\right)}^{N}=e^{-cr}=e^{-\tau }$$
where, the optical path length τ=cr, and the probability of interaction occurring within the optical path length τ, i.e., the cumulative probability distribution, is(5)$$\Psi (\tau )=1-e^{-\tau }$$

The optical path length is sampled as(6)τ=−ln⁡(1−ζ)

Since 1−ζ is also evenly distributed between 0 and 1, Equation (5) can also be written as(7)τ=−ln⁡(ζ)

If r is the transmission distance, then(8)$$r=\frac{\tau }{c}=-\frac{ln(\zeta )}{c}$$

#### 2.1.2. Types of Photon Interactions

After sampling the photon transmission distance r using the above random number method, the same method was used to determine whether the interaction type between photons and the water medium is absorption or scattering. Generate a random number ζ uniformly distributed from 0 to 1, and then compare it with the single scattering probability $\omega_{0}$. If ζ > ${\;\omega }_{0}$, it is an absorption interaction; If ζ ≤ ${\;\omega }_{0}$, it is a scattering interaction, and therefore ${\;\omega }_{0}$ also refers to the survival rate of photons. Assuming that the weight of the photon is equal to 1 at the beginning, the weight of the photon changes to ${\omega_{0}}^{n}$ after n times interactions with the medium. Additionally, boundary absorption and reflection were taken into account. To avoid invalid tracing and high computing costs, the tracking of the photon was interrupted when the photon weight was less than the threshold.

#### 2.1.3. Photon Scattering Direction

After the interaction between the photons and the medium, the new direction of photons is randomly determined based on the scattering phase function $\tilde{\beta }$. Therefore, the probability density of scattering into the unit solid angle $d\Omega (\vec{\xi })$) centered on direction $\vec{\xi }$ is(9)$$\tilde{\beta }\left(\vec{\xi^{\prime }}\to \vec{\xi }\right)d\Omega (\xi )=\tilde{\beta }(\theta ,\varphi )\sin\theta d\theta d\varphi$$
where θ and φ are the scattering angle and azimuth angle of photons in the coordinate system centered on the incident direction $\vec{\xi^{\prime }}$, respectively.

In natural water bodies, $\tilde{\beta }$ depends only on the scattering angle, θ and φ are independent random quantities that can be sampled separately using two random numbers. Therefore, the probability density function of the photon scattering direction could be written as(10)$$\tilde{\beta }(\theta ,\varphi )\sin\theta d\theta d\varphi =p_{\theta }(\theta )d\theta \cdot p_{\varphi }(\varphi )d\varphi$$

The azimuth angle φ is uniformly distributed within the range of 0~2π, the probability density function corresponding to φ is(11)$$p_{\varphi }(\varphi )d\varphi =(1/2\pi )d\varphi$$

Then, the random sampling of φ is(12)φ=2πζ
where ζ is another random variable uniformly distributed between 0 and 1.

The probability density function corresponding to θ is(13)$$p_{\theta }(\theta )=2\pi \tilde{\beta }(\theta )sin\theta$$

To determine the scattering angle θ, a random number ζ uniformly distributed between 0 and 1 was generated, and the cumulative distribution function $\Psi_{\theta }\left(\theta \right)$ is(14)$$\zeta =\Psi_{\theta }(\theta )=\int_{0}^{\theta }\;p_{\theta }\left(\theta^{\prime }\right)d\theta^{\prime }=2\pi \int_{0}^{\theta }\;\tilde{\beta }\left(\theta^{\prime }\right)\sin\theta^{\prime }d\theta^{\prime }$$

Due to the complex shape of $\tilde{\beta }(\theta )$, it is usually necessary to numerically solve Equation (14) to determine the θ corresponding to ζ.

Substituted the cosine of the scattering angle μ=cos⁡θ into Equation (14) could have(15)$$\zeta =\Psi_{\mu }(\mu )=\int_{\mu }^{1}\;p_{\mu }\left(\mu^{\prime }\right)d\mu^{\prime }=2\pi \int_{\mu }^{1}\;\tilde{\beta }\left(\mu^{\prime }\right)d\mu^{\prime }$$
where ζ is a random number evenly distributed between 0 and 1.

### 2.2. Simulation Settings

As mentioned above, the photon weight, transmission length, and scattering direction are determined by the optical properties of water. In this study, we focused on the three typical water types since they represent the global marine waters since the IOPs are obtained in the real sea [40,41], which are shown in Table 2 and Figure 1. The single scattering albedo (SSA) denoted by $\omega_{0}$ indicates the scattering strength in a certain water type. According to Table 2, it is evident that clear water is absorption-dominant ($\omega_{0}$ is below 0.5), coastal water has comparable absorption and scattering ($\omega_{0}$ is slightly higher than 0.5), and turbid harbor water is scattering-dominant ($\omega_{0}$ is higher than 0.5). When the scattering effect is strong in a particular water body, the transmitted laser beam will become significantly divergent. Moreover, there are differences in the SPFs of these three typical waters, with forward distinctions influencing the change in the scattering angle.

For the convenience of comparison, we utilized the attenuation length (=c·r, c is the attenuation coefficient of a water body) instead of the geometrical distance. The attenuation lengths of the coastal and turbid harbor water types ranged from 1 to 35. For clear water, the maximum attenuation length was set to 25. Effective results could not be produced in clear water when the attenuation length exceeded 25 due to the relatively high absorption (smaller $\omega_{0}$ value) and computational constraints.

In the simulation of this study, the light source was set to a parallel beam with a beam width of 2 mm and a wavelength of 532 nm. From the simulations, we obtained the critical parameters influencing the performance of the underwater laser communication system, such as the AOA distribution, beam distribution, and power loss. The AOA distribution refers to the variation of the normalized power per unit angle with angles relative to the axes perpendicular to the receiving plane, and beam distribution refers to the variation of the power loss per unit area on the receiving plane with an offset relative to the center of the receiving plane. To obtain the AOA distribution and beam distribution, plenty of receivers with full FOV were set on the whole receiving plane and placed in unequal intervals, which refers to the offsets used in subsequent analysis. Figure 2 exhibits how the photons transmit through the seawater channel and are then received by the receivers on the receiving plane.

## 3. Results and Discussion

### 3.1. Average AOA

To study the laser divergence characteristics, the average AOA was obtained from two sources: (a) the AOA distribution on the whole receiving plane (perpendicular to the incident laser) for analyzing the overall divergence characteristics; (b) the AOA distribution at different offsets relative to the beam center on the receiving plane for analyzing the laser divergence characteristics at specific offsets.

#### 3.1.1. AOA Distribution on the Whole Receiving Plane

Figure 3 shows the variations in the average AOA on the whole receiving plane for the different waters with the attenuation length. Generally, the average AOA of the various water types increases with increasing attenuation length. For example, when the attenuation length is 15, the average AOAs are 9.6°, 15.3°, and 26° in clear water, coastal water, and turbid harbor water, respectively. Compared to clear water, the variations in the average AOA are much higher in coastal and turbid harbor waters. For instance, when the attenuation length ranges from 1 to 25, the average AOA of clear water increases from 6.7° to 10.9° while it increases from 5.5° to 18° and from 7.7° to 28.1° in coastal and turbid harbor waters, respectively. As shown in Table 2, the single scattering albedos $\omega_{0}$ of the various waters significantly differ, with the minimum value obtained in clear water ($\omega_{0}=$0.25) and the highest value obtained in turbid harbor water ($\omega_{0}=$0.83), indicating that photons in turbid harbor water are scattered more than those in clear water at the same attenuation length, leading to the higher average AOA in turbid harbor water.

Notably, the average AOA in coastal water is lower than that in clear water at an attenuation length smaller than 3. According to the radiative transfer equation, the scattering angle is influenced by the $\omega_{0}$ and SPF. Specifically, a higher $\omega_{0}$ corresponds to a higher probability of scattering events, and a higher SPF suggests a greater possibility of specific scattering angles. To gain deeper insights into the effects of $\omega_{0}$ and SPF on the average AOA, we undertook a thorough quantitative analysis using parameter values from these three water types (detailed results are omitted). The findings show that while the impact of SPF diminishes as the attenuation length increases, the influence of $\omega_{0}$ remains robust. Therefore, the effect of scattering at smaller angles (less than 1°) may reach equilibrium with the $\omega_{0}$ effect at a certain attenuation length, as evidenced by the intersection of the black and red lines in Figure 3. Furthermore, turbid harbor water consistently demonstrates the highest average AOA among the three water types, primarily due to its significantly elevated $\omega_{0}$ value.

#### 3.1.2. Average AOA at Different Offsets on the Receiving Plane

To further analyze the laser divergence characteristics on the receiving plane at different offsets, AOA distributions at 0.01, 0.1, 1.0, 5, and 10 m offsets relative to the beam center were simulated. For simplicity, only the average AOA results are provided in this study. Figure 4 shows the variations in the average AOA with the attenuation length at the different offsets in clear water, coastal water, and turbid harbor water, respectively.

In absorption-dominated clear water, the average AOA decreases continuously as the attenuation length increases because of rapidly attenuating laser power. For instance, the average AOA of 10 m is approximately 63.4° at an attenuation length of 1, 15.2° at an attenuation length of 15, and 13.1° at an attenuation length of 25; the average AOAs at the offset of 0 m are 1.1°, 0.75°, and 0.15° at the attenuation lengths of 1, 15, and 25, respectively. Thus, the differences in the average AOAs between offsets 0 m and 10 m are 62.3°, 14.5°, and 12.9° at the above attenuation lengths. In scattering-dominated coastal and turbid waters, the average AOAs at large offsets decrease with increasing attenuation length and increase at small offsets since multiple scattering leads to spatial dispersion of the laser beam. Here are some examples. In coastal water, the average AOA at the offset of 10 m is 75.4°, 23.1°, and 20.2° at the attenuation lengths of 1, 20, and 35, respectively; the average AOAs at the offset of 0 m are 1.0°, 1.2°, and 14.4° at the attenuation lengths of 1, 20, and 35, respectively; the differences at these offsets are 74.4°, 21.9°, and 5.8° at the same attenuation lengths. In turbid harbor water, the average AOA at the offset of 10 m is 70.0°, 47.5°, and 39.7° at the attenuation lengths of 1, 20, and 35, respectively; the average AOAs at the offset of 0 m are 2.7°, 19.3°, and 27.4° at the attenuation lengths of 1, 20, and 35, respectively; the differences at these offsets are 67.2, 28.3°, and 12.2° at the same attenuation lengths. Overall, the differences in the average AOAs at various offsets are more significant at smaller attenuation lengths but decreased considerably at larger ones.

Figure 5 shows the AOA distribution with the attenuation length in clear water, coastal water, and turbid harbor water, demonstrating more specific characteristics of laser divergence. In summary, light concentrated at the small angles gradually diverges with increasing attenuation length, and the peak angle of the AOA distribution is gradually shifted toward larger angles. The higher the single scattering albedo $\omega_{0}$, the faster the laser diverges and the more divergent it becomes. For example, at an attenuation length of 15, the peak angle of the AOA distribution is close to 0°, whereas it is approximately 11.6° and 23° in coastal and turbid harbor waters, respectively. The peak angles of the AOA distributions at an attenuation length of 25 are approximately 8.5°, 15.6°, and 27°, with corresponding full width at half maxima (FWHMs) of 21.8°, 30.6°, and 45.4° in clear water, coastal water, and turbid harbor water, respectively. Furthermore, when the attenuation length exceeds 25, the AOA distributions in coastal and turbid harbor waters tend to remain constant. When the attenuation length is 35, the peak angles in coastal water and turbid harbor water are 16.6° and 27°, respectively, which is within a 1° increase relative to the peak angles at an attenuation length of 25. The sharp-peaked original laser beam is scattered after a certain attenuation length, and the final angular distribution only depends on the inherent optical characteristics (IOPs) of the water body following asymptotic radiative theory [4,43,44,45,46].

The analysis above highlights the crucial role of the AOA distribution in determining the FOV of the receiver and its impact on receiving gain. As shown in Figure 4, when the attenuation length is small, a high receiving gain could be obtained at the center of the receiving plane using a small FOV. However, the receiving gain rapidly decreases at offsets away from the center, so a large FOV is suitable. In addition, in coastal and turbid harbor waters, it is necessary to magnify the receiving field of view to obtain a high receiving gain at long attenuation lengths. These findings highlight the necessity of considering factors such as water type, transmission distance, and the position on the receiving plane when determining the FOV of the receiver.

### 3.2. Beam Spreading

Figure 6 shows the variation in the beam diameter on the receiving plane under different water properties and attenuation lengths. The beam diameter is twice the offset at which the laser power decays to 1% of the power at the center, calculated from beam distributions. With increasing attenuation length, the beam diameters in the various waters increase with distinct patterns. The beam diameter in clear water remained relatively stable, increasing from approximately 0.08 m at an attenuation length of 1 to 0.10 m at an attenuation length of 25. A small beam diameter is primarily due to the significant decline in laser power in clear water. In contrast, the changes in beam diameter are much more pronounced in coastal and turbid harbor waters, in which noticeable beam spreading occurs. In coastal water, the beam diameter is about 0.19 m at an attenuation length of 1 and increases gradually to 0.60 m at an attenuation length of 15. Then, it expands dramatically to 83.8 m when the attenuation length reaches 35. Similarly, in turbid harbor water, the beam diameter begins at about 0.18 m at an attenuation length of 1, increases to 8.9 m at an attenuation length of 15, and then rises to 25.3 m at an attenuation length of 35. It is important to note that laser energy attenuates more slowly in turbid harbor water due to high scattering ability, resulting in less significant beam spreading than in coastal water and a smaller final beam diameter in the harbor.

Figure 7 shows the laser beam distributions under different attenuation lengths in clear water, coastal water, and turbid harbor water, which provide a more detailed overview of the power attenuation process for the different offsets away from the beam center. Overall, at small attenuation lengths, the power rapidly decayed with increasing offset, but as the attenuation length increased, the power slowly declined. After reaching a certain attenuation length, the power loss remained uniform within the offset range of approximately 1 to 10 m in coastal and turbid waters. The higher $\omega_{0}$ is, the smaller the attenuation length. For example, the power loss remained constant at an offset of less than 1 m, with a corresponding attenuation length of 30 in coastal water and approximately 20 in turbid harbor water.

The analysis of underwater characteristics indicates that laser beam spreading significantly affects the receiving gain. For a given attenuation length, the characteristics of beam spreading differ among three typical types of water. In strongly scattering water bodies (e.g., coastal and turbid harbor waters), the beam spreading is noticeable, and the laser power remains consistent over a wide range of offsets at long attenuation lengths. In contrast, in weakly scattering water (e.g., clear water), the beam spreading is relatively inapparent, and the laser power decreases rapidly with increasing offset. Therefore, it is essential to ensure proper alignment between the transmitter and receiver to minimize power loss in clear water. Alignment is the requisite for high receiving gain at short attenuation lengths in coastal and turbid harbor waters, unnecessary at long attenuation lengths because linear misalignment of the receiver may not result in significant power loss at long attenuation lengths. It suggests the need to consider the type of water, transmission distance, and offset on the receiving plane when estimating receiving gain.

### 3.3. Channel Loss

To determine the variable features of the laser power with attenuation length at the various offsets, the channel loss (per unit area) was calculated as shown in Figure 8. For comparison, two theoretical laser power attenuation patterns $e^{-K_{d}r}$ and $e^{-cr}$ are also shown in Figure 8. The corresponding diffuse attenuation coefficients $K_{d}$ of the three typical waters are 0.120, 0.189, and 0.493 m^−1^, and the beam attenuation coefficients c are 0.151, 0.399, and 2.19 m^−1^, respectively. Beer’s law ($e^{-cr}$) governs the laser power attenuation at the center within a specific range of attenuation length. The pattern of laser power decline with attenuation length in clear water is strikingly identical to Beer’s law, as the black and blue dotted lines overlaid in Figure 8a demonstrate. By contrast, the pattern changes at attenuation lengths of 15 and 5 in coastal and turbid waters (the black and blue dotted lines in Figure 8b and Figure 8c, respectively), indicating that Beer’s law causes underestimation of the power received at the beam center at a large attenuation length [32]. The absolute differences in power loss between simulated results and Beer’s Law calculations could be almost 2 dB·m^−2^ and 8 dB·m^−2^ at the attenuation lengths of 1 and 15 in turbid harbor waters. Since the simulations only account for the forward scattering photons, the power losses of simulated results are invariably less than the values calculated from the pattern $e^{-K_{d}r}$.

For a specific offset, the characteristics of laser channel loss variation with attenuation length differ among water types. However, the absolute differences in power loss at various offsets reduce as the attenuation length increases. In weak scattering clear water, the power loss at the offset of 0 m increases rapidly with increasing attenuation length and the power losses are about 31.4 dB·m^−2^, −29.5 dB·m^−2^, and −74.4 dB·m^−2^ at the attenuation lengths of 1, 15, and 25; the power losses at the offset of 10 m are −56.6 dB·m^−2^, −84.8 dB·m^−2^, and −121.2 dB·m^−2^ at the same attenuation lengths. Thus, the absolute differences in the power between offsets 0 m and 10 m are 88.0 dB·m^−2^, 55.3 dB·m^−2^, and 46.8 dB·m^−2^ at the above attenuation lengths. In strong scattering waters, the power loss changes more slowly after reaching a certain attenuation length. In coastal water, the power losses at the offset of 0 m are 31.3 dB·m^−2^, −29.1 dB·m^−2^, and −106.2 dB·m^−2^ at the attenuation lengths of 1, 15, and 35, respectively; the power losses at the offset of 10 m are −63.6 dB·m^−2^, −63.6 dB·m^−2^, and −110.5 dB·m^−2^ at the same attenuation lengths; the absolute differences at these offsets are 94.9 dB·m^−2^, 34.5 dB·m^−2^, and 4.3 dB·m^−2^ at the same attenuation lengths. In turbid harbor water, the power losses at the offset of 0 m are 32.6 dB·m^−2^, −34.9 dB·m^−2^, and −58.6 dB·m^−2^ at the attenuation lengths of 1, 20, and 35, respectively; the power losses at the offset of 10 m are −84.8 dB·m^−2^, −60.5 dB·m^−2^, and −71.2 dB·m^−2^ at the same attenuation lengths; the absolute differences at these offsets are 117.4 dB·m^−2^, 25.6 dB·m^−2^, and 12.6 dB·m^−2^ at the same attenuation lengths. It indicates that power loss is significant in clear water and at short attenuation lengths in the two other waters due to the attenuation effect of the water body and less at long attenuation lengths in more turbid waters because of scattering contribution if there is a linear misalignment between the transmitter and the receiver [32]. Additionally, in coastal and turbid harbor waters, the power losses at offsets away from the center reduce within an attenuation length range and then increase when exceeding a certain attenuation length, noticeably in turbid harbor water, following similar changing laws in Figure 4.

The above analysis focuses on the characteristics of channel loss in underwater laser transmission. Channel loss has a significant impact on the strength of the received channel signal. The research indicates that channel loss is affected by water type, transmission distance, and the offset away from the center on the receiving plane. When the attenuation length increases, water bodies with strong scattering show a slower change in power loss compared to those with weak scattering. Additionally, the difference in channel energy loss at different offsets varies significantly across different water bodies and attenuation lengths.

## 4. Conclusions

Based on Monte Carlo ray tracing technology, the underwater laser transmission of a blue-green narrow beam was simulated. With the use of the average AOA, beam spreading, and channel loss, the laser transmission characteristics in three typical waters were analyzed.

The differences in the average AOAs at various offsets are more significant in clear water and at short attenuation lengths in coastal and turbid harbor waters because of power attenuation and then decrease considerably at long attenuation lengths in coastal and turbid harbor waters due to multiple scattering. For instance, in clear water, the difference in average AOA between offsets of 0 m and 10 m is approximately 62.3° at an attenuation length of 1, decreasing to 14.5° at an attenuation length of 15, and 12.9° at the attenuation length of 25. In coastal water, the differences in average AOAs are about 74.4°, 21.9°, and 5.8° at the attenuation lengths of 1, 20, and 35, respectively. In turbid harbor water, the differences in average AOAs are 67.2°, 28.3, and 12.2° at the attenuation lengths of 1, 20, and 35, respectively.

The characteristics of beam spreading in different water types are distinct. In clear water, the beam diameter is small, increasing from approximately 0.08 m to 0.10 m when the attenuation length changes from 1 to 25. In coastal water, the beam diameter increases gradually from 0.19 m to 0.60 m at an attenuation length of 1 and 15, then expands dramatically to 83.8 m at an attenuation length of 35. In turbid harbor water, the beam diameter is about 0.18 m at an attenuation length of 1 and 8.9 m at an attenuation length of 15 and then increases to 25.3 m at an attenuation length of 35.

The difference in the received power at the various offsets decreases with increasing attenuation length but with distinct patterns. Take the offsets of 0 m and 10 m as examples. In clear water, the absolute difference in the power loss is up to 88.0 dB·m^−2^ when the attenuation length is less than 10 and has no significant change exceeding the attenuation length of 10 with the absolute difference of 55.3 dB·m^−2^ at the attenuation length of 15 and 46.8 dB·m^−2^ at the attenuation length of 25. In coastal water, the absolute difference in the power loss has the maximum value of 94.9 dB·m^−2^ when the attenuation length is less than 10, then rapidly reduces to 34.5 dB·m^−2^ at the attenuation length of 15 and 4.3 dB·m^−2^ at the attenuation length of 35. In turbid harbor water, the maximum absolute difference in the power loss is 117.4 dB·m^−2^ when the attenuation length is less than 15, then gradually reduces to 25.6 dB·m^−2^ at the attenuation length of 20 and 12.6 dB·m^−2^ at the attenuation length of 35. Moreover, the minimum underestimation of power loss by applying Beer’s Law could be almost 2 dB·m^−2^ in turbid harbor waters.

Based on the above results, lasers exhibit different transmission characteristics in different water types. Therefore, when designing and developing UWOC systems, it is necessary to consider the water type in the application scenario. At short transmission distances, there is still a high receiving gain on the center of the receiving plane using a small FOV receiver, while a large FOV is required to achieve a high receiving gain at the positions away from the center. In coastal and turbid harbor waters, it is necessary to magnify the FOV of the receiver to achieve a high receiving gain at the long attenuation length. According to the results of the power losses at different offsets of the receiving plane in different waters, there are significant differences in the position strategies of receivers for different water types. In clear water, the beam spreading is inconspicuous, which indicates that the receiver must be at the center of the receiving plane to achieve high receiving gain. In coastal and turbid port waters, the receiver must be at the center of the receiving plane for high receiving gain at short transmission distances, but not necessarily at long transmission distances where the difference in received power at different positions on the receiving plane gradually decreases.

This study utilized the optical properties of three typical oceanic water types for simulations, leading to valuable insights for designing communication systems in various marine environments. This approach enables a comprehensive assessment of performance factors such as link distance, transmission rate, and error rate. This study takes the narrow beam laser wavelength of 532 nm as an example to analyze the transmission characteristics. However, there may be differences in transmission characteristics in other visible-light wavelengths due to differences in absorption and scattering properties. The wide-beam LED would result in differences in transmission characteristics as well. For this study, we considered three types of typical waters. Nevertheless, the real-time analysis of the transmission characteristics should account for the temporal and spatial changes in the properties of marine waters. Therefore, when designing communication systems for specific application scenarios, it is necessary to consider the light source used and the properties of the water type applied. In addition, other factors influencing the performance of the UWOC system, such as turbulence, upper-layer oceanic bubbles, and chlorophyll depth profile were not considered in this study. Future research should focus on analyzing temporal dispersion characteristics to enhance underwater communication solutions.

## Figures and Tables

**Figure 1 sensors-25-03057-f001:**
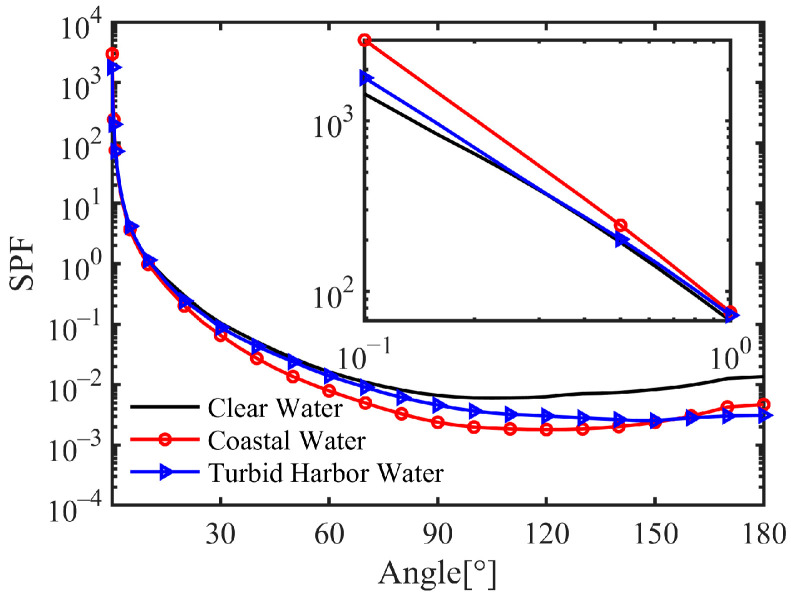
Scattering phase functions (SPFs) of the three typical waters [41].

**Figure 2 sensors-25-03057-f002:**
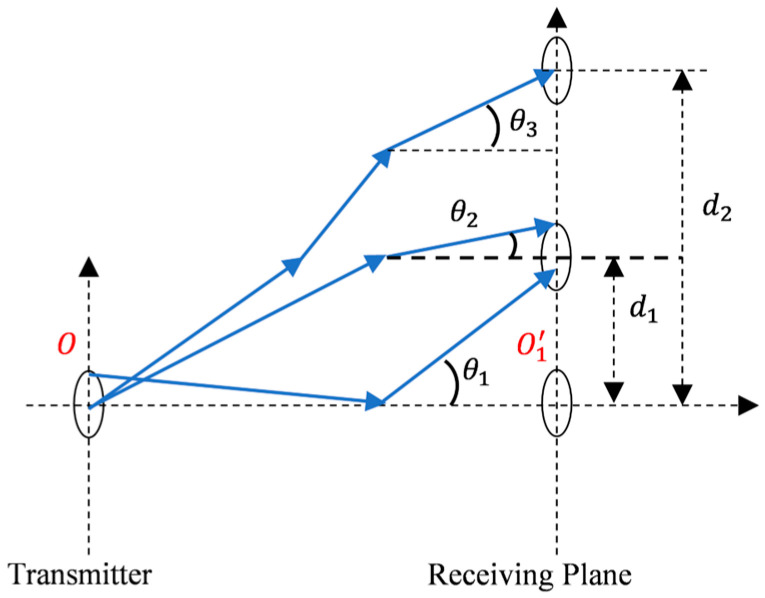
Receiving and transmitting schematic diagram.

**Figure 3 sensors-25-03057-f003:**
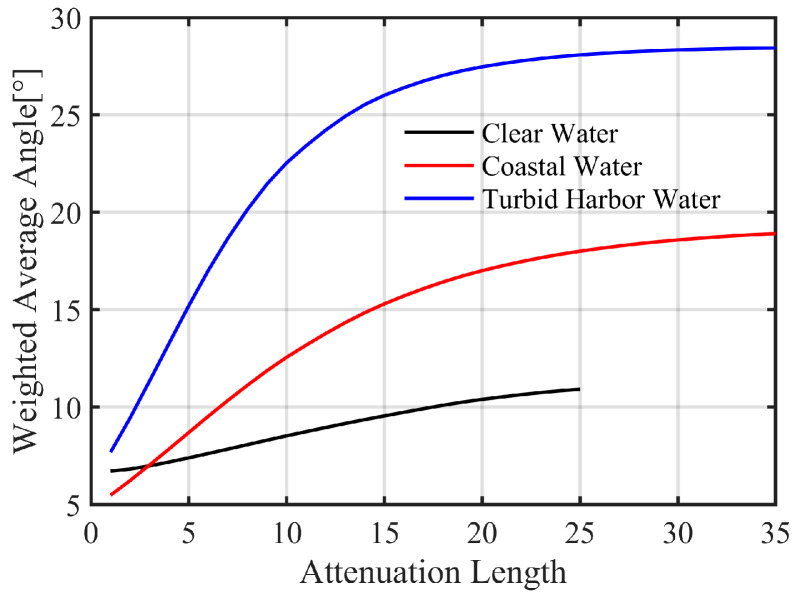
Variations in the average AOA with the attenuation length in clear water (black line), coastal water (red line), and turbid harbor water (blue line).

**Figure 4 sensors-25-03057-f004:**
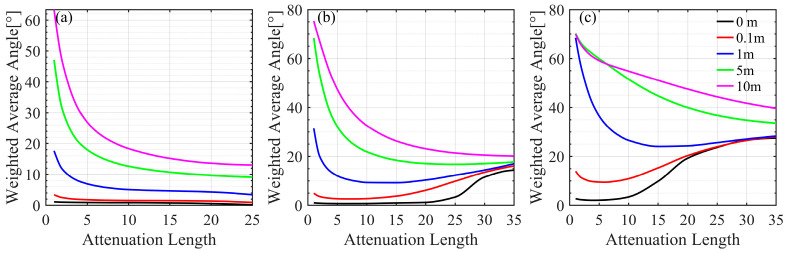
Variations in the average AOA with the attenuation length at the different offsets in (**a**) clear water, (**b**) coastal water, and (**c**) turbid harbor water.

**Figure 5 sensors-25-03057-f005:**
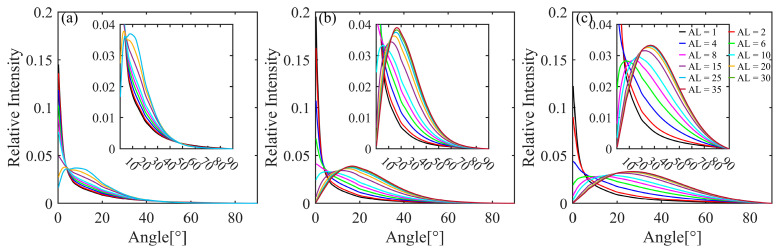
AOA distributions for the different attenuation lengths in (**a**) clear water, (**b**) coastal water, and (**c**) turbid harbor water.

**Figure 6 sensors-25-03057-f006:**
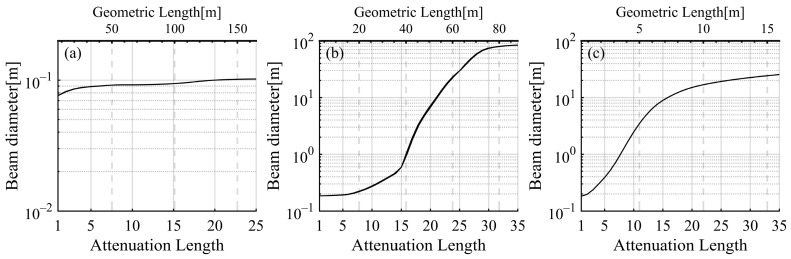
Variations in the beam diameter with attenuation length in (**a**) clear water, (**b**) coastal water, and (**c**) turbid harbor water.

**Figure 7 sensors-25-03057-f007:**
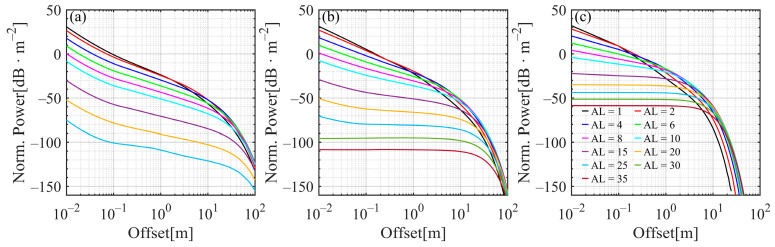
Beam distributions for the different attenuation lengths in (**a**) clear water, (**b**) coastal water, and (**c**) turbid harbor water.

**Figure 8 sensors-25-03057-f008:**
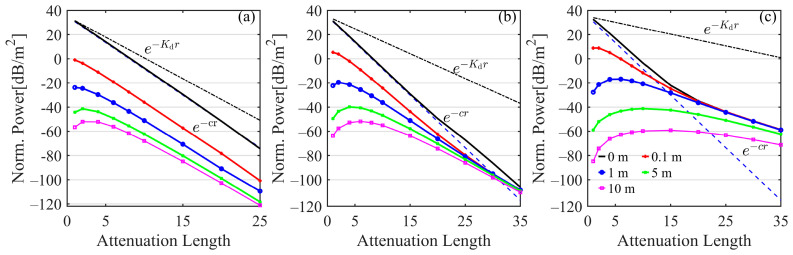
Variations in the normalized power loss with the attenuation length at the different offsets in (**a**) clear water, (**b**) coastal water, and (**c**) turbid harbor water (represented with solid lines); two theoretical channel loss models (i.e., $e^{-K_{d}r}$ and $e^{-cr}$) are drawn in dash-dotted and dashed lines, respectively.

**Table 1 sensors-25-03057-t001:** List of symbols.

Symbols	Descriptions	Unit
d	Offset relative to the beam center on the receiving plane	m
r	Transmission distance	m
a	Absorption coefficient	m^−1^
b	Scattering coefficient	m^−1^
c	Attenuation coefficient, c=a+b	m^−1^
τ	Optical path length, τ=cr	
ζ	Uniform random variable distributed between 0 and 1	
p(x)	Probability density distribution function of a certain sampling variable	
Ψ(x)	Cumulative distribution function of a certain sampling variable	
L	Radiance	W m^−2^ sr^−1^
θ	Scattering angle	°
φ	Azimuth angle	°
$\vec{\xi }$	Direction vector of a photon	
$\omega_{0}$	Single scattering albedo (SSA), $\omega_{0}=b/c$	
β	Volume scattering function (VSF)	m^−1^ sr^−1^
$\tilde{\beta }$	Scattering phase function (SPF), $\tilde{\beta }=\beta /b$	sr^−1^
$K_{d}$	Downwelling diffuse attenuation coefficient	m^−1^

**Table 2 sensors-25-03057-t002:** Optical properties of the three typical waters [42].

Water Type	a **(**$m^{-1}$**)**	b **(**$m^{-1}$**)**	c **(**$m^{-1}$**)**	$k_{d}$	$\omega_{0}$
Clear Water	0.114	0.037	0.151	0.120	0.25
Coastal Water	0.179	0.220	0.399	0.189	0.55
Turbid Harbor Water	0.366	1.829	2.195	0.493	0.83

## Data Availability

The raw data supporting the conclusions of this article will be made available by the authors on reasonable request.

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
