# Peer review of "Laser Transmission Characteristics of Seawater for Underwater Wireless Optical Communication"

_sensors, 2025, doi:10.3390/s25103057_

Round 1

Reviewer 1 Report

Comments and Suggestions for Authors

This study examines the transmission characteristics of underwater blue-green 
laser communication using the Monte Carlo photon tracing method. It is interesting to see the Monte Carlo approach for laser transmission characteristics. However, following concerns should be addressed to improve the quality.

1). The abstract of the paper should be improved. Main novelty is not mentioned. Clear idea about the methodology used should be included. Some important numerical findings also can be included. 

2). In addition to the literature presented in the paper, there are studies that consider inhomogeneous water conditions, and temporal dispersions. Better to study and refer them as well. Followings are some suggestions.

a). An Overview of Underwater Optical Wireless Communication Channel Simulations with a Focus on the Monte Carlo Method

b). Relay-Assisted Optical Wireless Communications in Turbid Water

3). The main contributions of the paper should be presented in the introduction  section.

4). The considered Monte Carlo simulation should be presented in the an algorithm.

5). Since there are large amount presented in the literature, what are the main novel concepts in the paper. Please highlight them carefully.

6). Do you consider losses due to misalignment of transmitter/receiver, reflections from air water surface or seabed, and blockages?

7). Reference format should be improved. Use abbreviations for journals. Use "in Proc. xx" for conferences. All references should be consistent.

Comments on the Quality of English Language

The use of professional English should be motivated. Check carefully for paragraph opening and clear explanations.

Author Response

Dear reviewer,

Thank you very much for taking the time to review this manuscript. Your opinions help to improve the academic rigor of our article. Based on your suggestion and request, we have made corrected modifications to the revised manuscript. We hope that our work can be improved again. In addition, we have made further revisions throughout the manuscript. The content of these major revisions is provided at the end of the letter.

Point-by-point response to Comments and Suggestions for Authors

Comments 1: [The abstract of the paper should be improved. Main novelty is not mentioned. Clear idea about the methodology used should be included. Some important numerical findings also can be included.]

Response 1: Thank you for this suggestion. According to the suggestion, the abstract has been rewritten to cover the main novelty, the methodology used, and the important numerical findings

Comments 2: [In addition to the literature presented in the paper, there are studies that consider

inhomogeneous water conditions, and temporal dispersions. Better to study and refer them as well. Followings are some suggestions. An Overview of Underwater Optical Wireless Communication Channel Simulations with a Focus on the Monte Carlo Method. Relay-Assisted Optical Wireless Communications in Turbid Water.]

Response 2: Thank you for your valuable suggestions. We have restructured Section 1 to integrate these key points. Inhomogeneous water conditions, temporal dispersions, and the analysis of laser application on relay-assisted systems are not considered in this study. Some relevant references are cited to support the discussion of channel modeling and system design, and development in paragraph 2 of Section 1, on page 2. The literature review concerning channel modeling using the Monte Carlo Method is presented in the revised manuscript. However, it focuses more on the transmission characteristics we have analyzed in the following chapters.

Comments 3: [The main contributions of the paper should be presented in the introduction section.]

Response 3: Thank you for this reminder. We have added the main contributions of this paper in paragraph 5 of Section 1, on page 3.

Comments 4: [The considered Monte Carlo simulation should be presented in the an algorithm.]

Response 4: Thank you for pointing this out. We add subsection 2.1 of Section 2 to introduce the Monte Carlo simulation method, including the main solution and the sampling of transmission distance, types of photon interactions, and scattering direction, as well as the associated probability density functions, with more details on pages 3 to 5.

Comments 5: [Since there are large amount presented in the literature, what are the main novel concepts in the paper. Please highlight them carefully.]

Response 5: Thank you for this suggestion. To highlight the main novel concepts, we have added descriptions in the introduction, results, and conclusion.

Comments 6: [Do you consider losses due to misalignment of transmitter/receiver, reflections from air water surface or seabed, and blockages?]

Response 6: Thank you for the valuable questions. Only the transmission process of laser in unbounded water bodies was taken into account in the simulations, without considering the power loss induced by the reflections from air, water, surface, or the seabed, and blockage. However, the program we have developed (Monte Carlo simulation) could be applied in channel loss evaluation on account of reflections from these factors. In this study, we have considered the losses due to the linear misalignment of the receiver. We have drawn the receiving and transmitting schematic diagram to explain this setting, as Figure 2 on page 6, in Subsection 2.2. We have modified the definition of the arrival angle distribution, the beam distribution, and the literal statement for Figure 2 to make it clearer.

“The AOA distribution refers to the variation of the normalized power per unit angle with angles relative to the axes perpendicular to the receiving plane, and beam distribution refers to the variation of the power loss per unit area on the receiving plane with an offset relative to the center of the receiving plane. To obtain the AOA distribution and beam distribution, plenty of receivers with full FOV are set on the whole receiving plane and placed in unequal intervals, which refers to the offsets used in subsequent analysis. Figure 2 exhibits how the photons transmit through the seawater channel and are then received by the receivers on the receiving plane.

Comments 7: [Reference format should be improved. Use abbreviations for journals. Use "in Proc. xx" for conferences. All references should be consistent.]

Response 7: Thank you for the reminder. We have checked the consistency of using abbreviations for journals, and the prefix "In Proc. " for conferences.

Response to Comments on the Quality of English Language

Point 1: The use of professional English should be motivated. Check carefully for paragraph opening and clear explanations.

Response 1: The manuscript has been polished and grammar checked, and we hope to present our research results more clearly in this revision.

There is a content of the major revisions:

1.      We have modified the abstract to present the novelty, significant quantitative results, and practical application of our study on page 1.

2.      Section 1 has been arranged in 6 paragraphs on pages 1 to 3. Thank you for this instructive suggestion. We have rearranged Section 1 into six paragraphs. Paragraph 1 introduces the importance of developing underwater laser communication; paragraphs 2, 3, and 4 are the literature reviews with the topic of channel modeling considering the water properties influence, the advantages of using the Monte Carlo method for underwater channel simulation, and the transmission characteristics research, separately; paragraph 5 introduces the main contribution of this study; paragraph 6 is a closing paragraph listing the paper structure. To specify the simulation method used in this study, "photon tracing" has changed to "ray tracing".

3.      Table 1 has been renamed to "List of symbols" and moved to Section 2 on page 3. In addition, a column of the unit has been added. The descriptions of the symbols used in Equation (1) introduced in Section 1 have been removed. The List of abbreviations used in this study is attached on page 13.

4.      Section 2 has been modified and separated into two subsections on pages 3 to 7. Subsection 2.1 is an introduction to the Monte Carlo ray tracing method on pages 3 to 5, and subsection 2.2 is about the simulation settings on pages 5 to 7.

5.      Revised part in Section 3 on pages 8 to 11. (1) To introduce a practical use of weighted average angle of arrival, the paragraph concerned with the data analysis has been modified on page 8; the angle of arrival distributions for the different attenuation lengths are given in Figure 5 and the associated analysis has been supplied on page 8 to 9; and a practical use of the angle of arrival distributions is described on page 9; (2) the paragraph concerned with the data analysis on the beam diameter has been modified on page 9; the beam distributions for the different attenuation lengths are given in Figure 7 and the associated analysis has been supplied on page 10; and practical use of the beam distributions is described on page 10; (3) The quantitative analysis of power loss has been added and the mistakes of the y-tick labels for Figure 8(a) and the figure annotation have been corrected on page 11.

6.      In Section 4. We have added considerations for significant quantitative results, application scenario system design, and the limits of this research.

Reviewer 2 Report

Comments and Suggestions for Authors

See attached file.

Author Response

Dear reviewer,

Thank you very much for your comments and professional advice. These opinions help to improve the academic rigor of our article. Based on your suggestion and request, we have made corrected modifications to the revised manuscript. We hope that our work can be improved again. In addition, we have made further revisions throughout the manuscript. The content of these major revisions is provided at the end of the letter.

Comments 1: [References. Some of them are too old (i.e., 1, 3, 4, 7, 8, 9, 10, 11, 12, 13, 14, 15, 16, 17, 19, and 20). Please, consider substituting them with similar contributions published from 2019 on, or alternatively provide reasons to keep them.]

Response 1: Thank you for pointing this out. We have updated the majority of them to the publish date from 2019 till now, and kept some references because of their instructive meaning, and we have not found the alternatives yet.

Comments 2: [Abstract. Please, give some hints about the most significant quantitative results obtained.]

Response 2: Thanks for your advice. We have added the significant quantitative results in the abstract.

Comments 3: [Section 1. In order to enhance the paper readability, I strongly suggest to split this Section into two. The former should only deal with contextualising the paper, presenting the problem to be solved, and listing the paper contributions. The latter should only deal with related works, by clearly presenting them and comparing them with this paper, especially by highlighting similarities and discrepancies. Moreover, the latter Section must clearly state how this work advances the current state-of-the-art about the topic.]

Response 3: Thank you for this instructive suggestion. We have rearranged Section 1 into six paragraphs. Paragraph 1 introduces the importance of developing underwater laser communication; paragraphs 2, 3, and 4 are the literature reviews with the topic of channel modeling considering the water properties influence, the advantages of using the Monte Carlo method for underwater channel simulation, and the transmission characteristics research, separately; paragraph 5 introduces the main contribution of this study; paragraph 6 is a closing paragraph listing the paper structure.

Comments 4: [Related Works Section. In order to provide readers with a broader perspective about the tackled topic, I suggest the Authors to include the following references [1, 2, 3, 4]. However, I also strongly encourage the Authors to perform additional research.]

Response 4: We appreciate you providing these references. The references have been adopted in this revision.

Comments 5: [Section 1. The Section misses of a closing paragraph listing the paper structure.]

Response 5: Thank you for this reminder. We have supplied the closing paragraph of Section 1 to list the paper structure on page 3.

Comments 6: [Section 1. The Section misses of a closing paragraph listing the paper structure.]

Response 6: Thank you for this feedback. The symbol c is widely used to present it in the ocean optics area and also the studies concerning the influence of water types on underwater channels (Gabriel et al. 2013; Elamassie et al. 2019; Huang et al. 2020; Sahoo et al. 2022; Mobley et al. 2022). Therefore, to be consistent with the widely accepted convention, we still use c to represent the beam attenuation coefficient in this study. In addition, we have presented the list of symbols in Table 1 in Section 2 to specify these descriptions, on page 3..

Comments 7: [Figures. All of the Figures having the attenuation length on the x-axis miss of the relative unit of measurement.]

Response 7: We appreciate the reviewer’s suggestion regarding the correctness of the unit for the attenuation length. However, the attenuation length is a dimensionless variable, which is the product of the attenuation coefficient (unit: m) and the transmission distance (unit: m-1). We have listed these parameters in Table 1 without specifying the unit in the original manuscript. This time we have added the column of unit to clarify in Table 1, on page 3.

Comments 8: [Details about the probability distributions of the stochastic elements involved in the simulations are missing.]

Response 8: Thank you for pointing this out. We have added the subsection of Monte Carlo simulation in Section 2 to cover the probability distributions of sampling variables, see pages 4 to 6.

Comments 9: [Simulation results must be presented by showing their distributions (box-plots may definitely hep), owing to the stochasticity of the simulated phenomenon.]

Response 9: Thank you for this advice. Although the Monte Carlo simulation process is stochastic, we used a sufficient number of photons (up to 1010 on average) in the simulation process to remove the stochastic influence, thus normalization and interpolation are used in data processing rather than "box plots".

Comments 10: [The presented results must be extensively discussed more in detail.]

Response 10: Thank you for this suggestion. We have restructured Section 3 by supplying the original distributions of the distribution of the angle of arrival as Figure 5 and the beam distribution as Figure 7, and the analysis is attached above the figures. We have reanalyzed the weighted average angle of arrival, the beam diameter, and the power loss results to point out significant quantitative results. Changes are on pages 7 to 11.

Comments 11: [I deem that readers will notably benefit if the Authors provide a series of practical examples in which the proposed system may find room in field activities, in order to give a more practical perspective to the paper.]

Response 11: We agree with this suggestion. We have considered the influence of water type, transmission distance, and offset relative to the center of the receiving plane on the angle of arrival, beam spreading, and power loss in Section 3. Further, the determination of field-of-view and the necessity of alignment for circumstances considering different water types and transmission distances are presented in paragraph 5 of Section 4, on page 12.

Comments 12: [Section 4. Please, resume the most significant quantitative obtained results.]

Response 12: Thank you for pointing this out. We have added the significant quantitative results to the conclusions in paragraphs 2 to 4 of Section 4, on page 12.

Comments 13: [Finally, the Authors must clearly state the limitations of the proposed approach.]

Response 13: Thank you for this suggestion. We have added the limitations of our study in paragraph 6 of Section 4, on pages 12 to 13. This study has not considered the influence of the light source and other underwater factors, and matched temporal dispersion could be simulated and analyzed in future works.

Here is the content of the major revisions:

  1. We have modified the abstract to present the novelty, significant quantitative results, and practical application of our study on page 1.
  2. Section 1 has been arranged in 6 paragraphs on pages 1 to 3. Thank you for this instructive suggestion. We have rearranged Section 1 into six paragraphs. Paragraph 1 introduces the importance of developing underwater laser communication; paragraphs 2, 3, and 4 are the literature reviews with the topic of channel modeling considering the water properties influence, the advantages of using the Monte Carlo method for underwater channel simulation, and the transmission characteristics research, separately; paragraph 5 introduces the main contribution of this study; paragraph 6 is a closing paragraph listing the paper structure. To specify the simulation method used in this study, "photon tracing" has changed to "ray tracing".
  3. Table 1 has been renamed to "List of symbols" and moved to Section 2 on page 3. In addition, a column of the unit has been added. The descriptions of the symbols used in Equation (1) introduced in Section 1 have been removed. The List of abbreviations used in this study is attached on page 13.
  4. Section 2 has been modified and separated into two subsections on pages 3 to 7. Subsection 2.1 is an introduction to the Monte Carlo ray tracing method on pages 3 to 5, and subsection 2.2 is about the simulation settings on pages 5 to 7.
  5. Revised part in Section 3 on pages 8 to 11. (1) To introduce a practical use of weighted average angle of arrival, the paragraph concerned with the data analysis has been modified on page 8; the angle of arrival distributions for the different attenuation lengths are given in Figure 5 and the associated analysis has been supplied on page 8 to 9; and a practical use of the angle of arrival distributions is described on page 9; (2) the paragraph concerned with the data analysis on the beam diameter has been modified on page 9; the beam distributions for the different attenuation lengths are given in Figure 7 and the associated analysis has been supplied on page 10; and practical use of the beam distributions is described on page 10; (3) The quantitative analysis of power loss has been added and the mistakes of the y-tick labels for Figure 8(a) and the figure annotation have been corrected on page 11.
  6. In Section 4. We have added considerations for significant quantitative results, application scenario system design, and limitations of this research.

Reviewer 3 Report

Comments and Suggestions for Authors

This study examines the transmission characteristics of underwater blue-green laser communication using the Monte Carlo photon tracing method. It focuses on crucial parameters such as the angle of arrival (AOA), beam spreading, and channel loss across three different water bodies. The following concerns could be considered to improve the current version.

1. The "Monte Carlo photon tracing method" is used in this work, however, its basic theory is not described in the paper. The expemental theory and method should be presented in Section 2. 
2. The blue-green laser is used in the analysis, however, the conclusions are summarized for underwater optical Wireless communication, I think it is not appropriate. Are the conclusions suitable for the semiconductor white laser, monochromatic laser or the LED white light as the light source? 
3. The specific exprimental enviroment, platform, parameters should be introduced in Section 3. Frankly, I don't know what the specific Underwater Optical Wireless Communication system looks like, maybe the authors add it in the introduction section this paper.
4. I am not familiar with the underwater wireless communication, but I think the distance of the the underwater wireless communication is short. The demand for long-distance and high-speed underwater data transfer should be wire communication. Maybe the blue-green laser is suitable for underwater wireless communication. However, it is not described in section 1.
5. The references in this article are outdated, the authors have to add the state-of-the art references in the manuscript.

Author Response

Dear reviewer,

Thank you very much for taking the time to review this manuscript. These opinions help to improve the academic rigor of our article. Based on your suggestion and request, we have made corrected modifications to the revised manuscript. We hope that our work can be improved again. In addition, we have made further revisions throughout the manuscript. The content of these major revisions is provided at the end of the letter.

Comments 1: [The "Monte Carlo photon tracing method" is used in this work, however, its basic theory is not described in the paper. The expemental theory and method should be presented in Section 2.]

Response 1: Thank you for pointing this out. We have supplied subsection 2.2 about the Monte Carlo ray tracing method, on pages 3 to 5.

Comments 2: [The blue-green laser is used in the analysis, however, the conclusions are summarized for underwater optical Wireless communication, I think it is not appropriate. Are the conclusions suitable for the semiconductor white laser, monochromatic laser or the LED white light as the light source?]

Response 2: Thanks for your valuable question. We have used the narrow beam with a beamwidth of 2 mm and a wavelength of 532 nm in the simulations, so the result is suitable for a monochromatic laser. We have put the lack of analysis of the influence of light source as one of the limitations of our study in paragraph 6 of Section 4, on pages 12 to 13.

Comments 3: [The specific exprimental enviroment, platform, parameters should be introduced in Section 3. Frankly, I don't know what the specific Underwater Optical Wireless Communication system looks like, maybe the authors add it in the introduction section this paper.]

Response 3: Thank you for this feedback. This study is about the laser transmission characteristics in different marine water bodies, aiming to provide basic support for the design and optimization of underwater laser communication systems in different application scenarios, without involving specific platforms and parameters.

Comments 4: [I am not familiar with the underwater wireless communication, but I think the distance of the the underwater wireless communication is short. The demand for long-distance and high-speed underwater data transfer should be wire communication. Maybe the blue-green laser is suitable for underwater wireless communication. However, it is not described in section 1.]

Response 4: Thank you for this valuable suggestion. We have modified this statement in Section 1. In general, the distance of underwater wireless optical communication could achieve short- to medium-distance applications (usually tens of meters), and its advantage is a high transmission rate compared to acoustic communication.

Comments 5: [Since there are large amount presented in the literature, what are the main novel concepts in the paper. Please highlight them carefully.]

Response 5: Thank you for pointing this out. We have updated the references mainly in Section 1 to the year from 2019 till now, while reserving some instructive references.

There is a content of the major revisions:

1.      We have modified the abstract to present the novelty, significant quantitative results, and practical application of our study on page 1.

2.      Section 1 has been arranged in 6 paragraphs on pages 1 to 3. Thank you for this instructive suggestion. We have rearranged Section 1 into six paragraphs. Paragraph 1 introduces the importance of developing underwater laser communication; paragraphs 2, 3, and 4 are the literature reviews with the topic of channel modeling considering the water properties influence, the advantages of using the Monte Carlo method for underwater channel simulation, and the transmission characteristics research, separately; paragraph 5 introduces the main contribution of this study; paragraph 6 is a closing paragraph listing the paper structure. To specify the simulation method used in this study, "photon tracing" has changed to "ray tracing".

3.      Table 1 has been renamed to "List of symbols" and moved to Section 2 on page 3. In addition, a column of the unit has been added. The descriptions of the symbols used in Equation (1) introduced in Section 1 have been removed. The List of abbreviations used in this study is attached on page 13.

4.      Section 2 has been modified and separated into two subsections on pages 3 to 7. Subsection 2.1 is an introduction to the Monte Carlo ray tracing method on pages 3 to 5, and subsection 2.2 is about the simulation settings on pages 5 to 7.

5.      Revised part in Section 3 on pages 8 to 11. (1) To introduce a practical use of weighted average angle of arrival, the paragraph concerned with the data analysis has been modified on page 8; the angle of arrival distributions for the different attenuation lengths are given in Figure 5 and the associated analysis has been supplied on page 8 to 9; and a practical use of the angle of arrival distributions is described on page 9; (2) the paragraph concerned with the data analysis on the beam diameter has been modified on page 9; the beam distributions for the different attenuation lengths are given in Figure 7 and the associated analysis has been supplied on page 10; and practical use of the beam distributions is described on page 10; (3) The quantitative analysis of power loss has been added and the mistakes of the y-tick labels for Figure 8(a) and the figure annotation have been corrected on page 11.

6.      In Section 4. We have added considerations for significant quantitative results, application scenario system design, and the limits of this research

Round 2

Reviewer 1 Report

Comments and Suggestions for Authors

Authors have well addressed all the comments. No further comments from the reviewer.

Reviewer 2 Report

Comments and Suggestions for Authors

The paper notably improved after its revision, and I have no further comments.

Reviewer 3 Report

Comments and Suggestions for Authors

Thanks for the revisions.